# Aliasing coincides with CNNs vulnerability towards adversarial attacks

**Julia Grabinski[1], Janis Keuper[3,4], Margret Keuper[1,2]**

[1]University of Siegen, Germany,
[2]Max Planck Institute for Informatics, Saarland Informatics Campus, Germany
[3] Institute for Machine Learning and Analytics (IMLA), Offenburg University, Germany
[4]Competence Center High PerformanceComputing, Fraunhofer ITWM, Kaiserslautern, Germany

## Abstract

Many commonly well-performing convolutional neural network models have shown to be susceptible to input data perturbations, indicating a low model robustness. Adversarial attacks are thereby specifically optimized to reveal model weaknesses, by generating small, barely perceivable image perturbations that flip the model prediction. Robustness against attacks can be gained for example by using adversarial examples during training, which effectively reduces the measurable model attackability. In contrast, research on analyzing the source of a model's vulnerability is scarce. In this paper, we analyze adversarially trained, robust models in the context of a specifically suspicious network operation, the downsampling layer, and provide evidence that robust models have learned to downsample more accurately and suffer significantly less from aliasing than baseline models.

## Introduction

Convolutional Neural Networks (CNNs) provide highly accurate predictions in a wide range of applications. Yet, to allow for practical applicability, CNN models should not be fooled by small image perturbations, as they are realized by adversarial attacks (Goodfellow, Shlens, and Szegedy 2015a; Moosavi-Dezfooli, Fawzi, and Frossard 2016; Rony et al. 2019). Such attacks aim to fool the network by perturbing image pixels such that human observers would still easily recognize the correct class label, while the network makes incorrect predictions. Susceptibility to such perturbations is prohibitive for the applicability of CNN models in real-world scenarios, as it indicates limited reliability and generalization of the model.

To establish adversarial robustness many sophisticated methods have been developed (Goodfellow, Shlens, and Szegedy 2015a; Rony et al. 2019; Kurakin, Goodfellow, and Bengio 2017; Goodfellow, Shlens, and Szegedy 2015b). Some can defend only against one specific attack (Goodfellow, Shlens, and Szegedy 2015a) while others propose more general defences against diverse attacks. Another way to protect CNNs against adversarial examples is to detect them. Harder et al. (2021) classify adversarial examples by inspecting each input image and its feature maps in the fre-

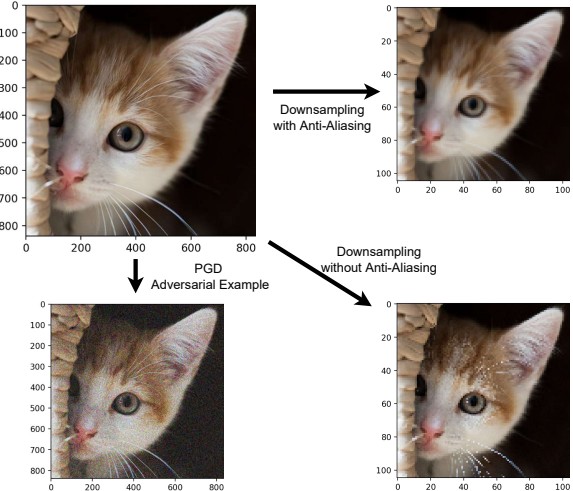

Figure 1: Illustration of down-sampling, with (top right) and without anti-aliasing filter (bottom right) as well as an adversarial example (bottom left). The top left image shows the original, on the top right, this image is correctly downsampled with an anti-aliasing filter. In the bottom right, no filter is applied, leading to aliasing. The adversarial example (bottom left) shows visually similar artifacts. In this paper, we investigate the role of aliasing for adversarial robustness.

quency domain. Similarly, Yin et al. (2020) showed that natural images and adversarial examples differ significantly in their frequency spectra.

In fact, when considering the architecture of commonly employed CNN models, one could wonder why these models perform so well although they ignore basic sampling theoretic foundations. Concretely, most architectures subsample feature maps without ensuring to sample above the Nyquist rate (Shannon 1949), such that, after each downsampling operation, spectra of sub-sampled feature maps may overlap with their replica. This is called *aliasing* and implies that the network should be genuinely unable to fully restore an image from its feature maps. One can only hypothesize that common CNNs learn to (partially) compensate for this effect by learning appropriate filters. Following this line of thought, recently, several works suggest im-

proving CNNs by including anti-aliasing techniques during down-sampling in CNNs (Zhang 2019; Zou et al. 2020). They aim to make the models more robust against image-translations, such that the class prediction does not suffer from small vertical or horizontal shifts of the content.

In this paper, we further investigate the relationship between adversarial robustness and aliases. While previous works (Yin et al. 2020; Harder et al. 2021; Lorenz et al. 2021) focused on adversarial examples, we systematically analyze potential aliasing effects inside CNNs. Specifically, we compare several recently proposed adversarially robust models to models which result from conventional training schemes in terms of aliasing. We inspect intermediate feature maps before and after the down-sampling operation at inference. Our first observation is that these models indeed fail to sub-sample according to the Nyquist Shannon Theorem (Shannon 1949): we observe severe aliasing. Further, our experiments reveal that adversarially trained networks exhibit less aliasing than standard trained networks, indicating that adversarial training encourages CNNs to learn how to properly down-sample data without severe artifacts.

In summary, our contributions are:

- We introduce a measure for aliasing and show that common CNN down-sampling layers fail to sub-sample the feature maps in a Nyquist-Shannon conform way.
- We analyze various adversarially trained models, that are robust against a strong ensemble of adversarial attacks, AutoAttack (Croce and Hein 2020), and show that they exhibit significantly less aliasing than standard models.

## Aliasing in CNNs

CNNs usually have a pyramidal structure in which the data is progressively sub-sampled in order to aggregate spatial information while the number of channels increases. During sub-sampling, no explicit precautions are taken to avoid aliases, which arise from under-sampling. Specifically, when sub-sampling with stride 2, any frequency larger than $N/2$, where $N$ is the size of the original data, will cause pathological overlaps in the frequency spectra. Those overlaps in the frequency spectra cause ambiguities such that high frequency components appear as low frequency components. Hence, local image perturbations can become indistinguishable from global manipulations.

### Aliasing Metric

To measure the possible amount of aliasing appearing after down-sampling we compare each down-sampled feature map in the Fourier domain with its aliasing-free counterpart. To this end, we consider a feature map $f(x)$ of size $2N \times 2N$ before down-sampling. We compute an "aliasing-free" down-sampling by extracting the $N$ lowest frequencies along both axes in Fourier space. W.l.o.G., we consider specifically down-sampling operations by strided convolutions, since these are predominantly used in adversarially robust models (Zagoruyko and Komodakis 2017).

In each strided convolution, the input feature map $f(x)$ is convolved with the learned weights $w$ and downsampled by strides, thus potentially introducing frequency replica (i.e. aliases) in the downsampled signal $\hat{f}_{s2}$.

$$\hat{f}_{s2} = f(x) * g(w, 2) \tag{1}$$

To measure the amount of aliasing, we explicitly construct feature map frequency representations without such aliases. Therefore, the original feature map $f(x)$ is convolved with the learned weights $w$ of the strided convolution without applying the stride $g(w, 1)$ to obtain $\hat{f}_{s1}$.

$$\hat{f}_{s1} = f(x) * g(w, 1) \tag{2}$$

Afterwards the 2D FFT of the new feature maps $\hat{f}_{s2}$ is computed, which we denote $F_{s2}$.

$$F_{s2}(k, l) = \frac{1}{N^2} \sum_{m=0}^{N-1} \sum_{n=0}^{N-1} \hat{f}_{s2}(m, n) e^{-2\pi j(\frac{k}{M}m + \frac{l}{N}n)}, \tag{3}$$

for $k, l = 0, \ldots, N - 1$. For the non-down-sampled feature maps $\hat{f}_{s1}$, we proceed similarly and compute for $k, l = 0, \ldots, 2 \cdot N - 1$

$$F_{s1}^{\uparrow}(k, l) = \frac{1}{4N^2} \sum_{m=0}^{2N-1} \sum_{n=0}^{2N-1} \hat{f}_{s1}(m, n) e^{-2\pi j(\frac{k}{2M}m + \frac{l}{2N}n)}. \tag{4}$$

The aliasing free version $F_{s1}$ can be obtained by setting all frequencies above the Nyquist rate to zero before down-sampling,

$$F_{s1}^{\uparrow}(k, l) = 0 \tag{5}$$

for $k \in [N/2, 3N/2]$ and for $l \in\in [N/2, 3N/2]$. Then the down-sampled version in the frequency domain corresponds to extracting the four corners of $F_{s1}^{\uparrow}$ and reassembling them as shown in Figure 2,

$$
\begin{aligned}
F_{s1}(k, l) &= F_{s1}^{\uparrow}(k, l) && \text{for} \quad k, l = 0, \ldots, N/2 \\
F_{s1}(k, l) &= F_{s1}^{\uparrow}(k + N, l) && \text{for} \quad k = N/2, \ldots, N \\
& && \text{and} \quad l = 0, \ldots, N/2 \\
F_{s1}(k, l) &= F_{s1}^{\uparrow}(k, l + N) && \text{for} \quad k = 0, \ldots, N/2 \\
& && \text{and} \quad l = N/2, \ldots, N \\
F_{s1}(k, l) &= F_{s1}^{\uparrow}(k + N, l + N) && \text{for} \quad k, l = N/2, \ldots, N
\end{aligned} \tag{6}
$$

This way we guarantee that there are no overlaps, i.e. aliases, in the frequency spectra. Figure 2 illustrates the computing process of the aliasing free down-sampling in the frequency domain. The aliasing free feature map can be compared to the actual feature map in the frequency domain to measure the degree of aliasing. The full procedure is shown in Figure 3, where we start on the left with the original feature map. Then we obtain the two down-sampled versions (with and without aliases) and get the difference between both by taking the $L_1$ norm.

The overall aliasing metric $AM$ for a down-sampling operation is calculated by taking the $L_1$ distance between downsampled and alias-free feature maps $f_k$ in the Fourier domain, averaged over K generated feature maps,

$$AM = \frac{1}{K} \sum_{k=0}^{K} |F_{s1,k} - F_{s2,k}|. \tag{7}$$

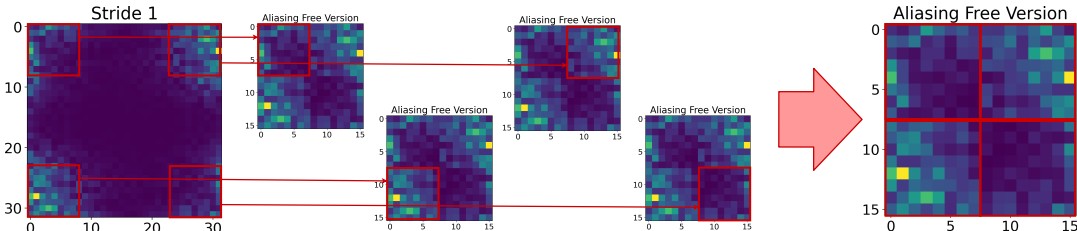

Figure 2: Step by step computation of the aliasing free version of a feature map. The left image shows the magnitude of the Fourier representation of a feature map with the zero-frequency in the upper left corner, i.e. high frequencies are in the center. Alias-free downsampling suppresses high frequencies prior to sampling. This can be implemented efficiently in the Fourier domain by cropping and reassembling the low-frequency regions of the Fourier representations, i.e. its four corners. Aliasing would correspond to folding the deleted high frequency components into the constructed representation.

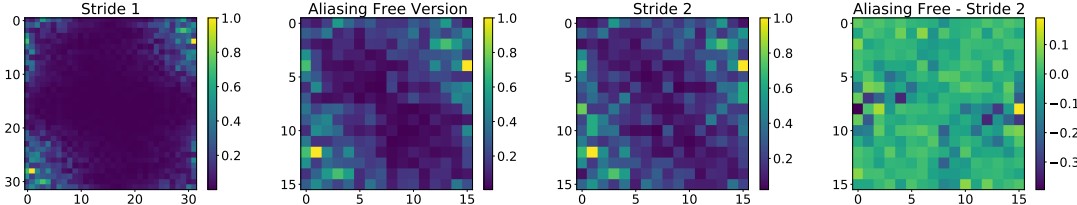

Figure 3: FFT (Fast Fourier Transformation) of a feature map in the original resolution (left). This feature map is downsampled by striding with a factor of two after aliasing suppression (middle left) and with aliasing (middle right). The difference between the original and aliasing-free FFT of the down-sampled feature map (right).

The proposed $AM$ measure is zero if aliasing is visible in none of the down-sampled feature maps, i.e. if sampling has been performed above the Nyquist rate. Whenever $AM$ is greater than 0, this is not the case and we should, from a theoretic point of view, expect the model to be easy to attack since it can not reliably distinguish between fine details and coarse input structures.

## Experiments

We conducted an extensive analysis of already existing adversarially robust models trained on CIFAR-10 (Krizhevsky 2012) with two different architectures, namely WideResNet-28-10 (WRN-28-10) (Zagoruyko and Komodakis 2017) and Preact ResNet-18 (He et al. 2016). Both architectures are commonly supported by many adversarial training approaches. As baseline, we trained a plain WRN-28-10 and Preact ResNet-18, both with similar training schemes. All adversarially trained networks are pre-trained models provided by RobustBench (Croce et al. 2020).
The WRN-28-10 networks have four operations in which down-sampling is performed. These operations are located in the second and third block of the network. In comparison, the Preact ResNet-18 networks have six down-sampling operations, located in the second, third and fourth layers of the network.

Both architectures have similar building blocks and the key operations including down-sampling are shown abstractly in the appendix in Figure 6. Each block starts with a convolution with stride two followed by additional operations like ReLu and convolutions with stride one. The char-

acteristic skip connection of ResNet architectures also needs to be implemented with stride two if down-sampling is applied in the according block. Consequently, we need to analyze all down-sampling units and skip connections before they are summed up to form the output feature map.

**WideResNet 28-10** In the following, differently trained WRN-28-10 networks are compared in terms of their robust accuracy against AutoAttack (Croce and Hein 2020) and the amount of aliasing in their down-sampling layer. The training procedure of the baseline can be found in the appendix.

Figure 4 indicates significant differences between adversarially trained and standard trained networks. First, the standard trained networks are not able to reach any robust accuracy, meaning their accuracy under adversarial attacks is equal to zero. Second, and this is most interesting for our investigation, standard trained networks exhibit much more aliasing during their down-sampling layer than adversarially trained networks. Through all layers and operations in which down-sampling is applied, the adversarially trained networks (blue dots) have much higher robust accuracy and much less aliasing compared to the standard trained networks. Additionally, we can observe that the amount of aliasing in the second layer is much higher than in the third layer. This can be explained by the different feature map sizes in the two layers as we calculate the absolute $L_1$ norm.

When comparing the conventionally trained network against each other it can be seen that also the specific training scheme used for training the network can have an influence on the amount of aliasing of the network. Concretely, the standard baseline model provided by Robust-

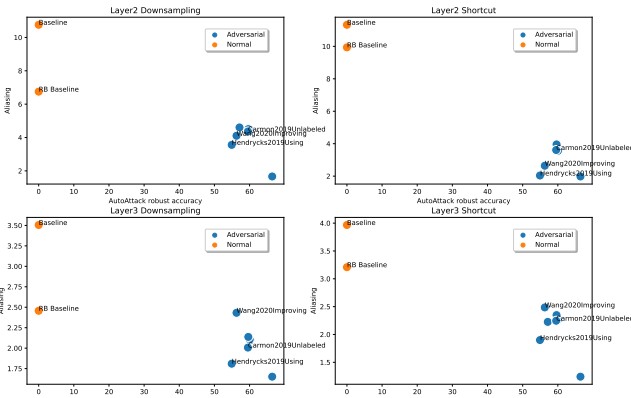

Figure 4: Adversarial Robustness versus Aliasing exemplary evaluated on different pre-trained WRN-28-10 models from RobustBench (Croce et al. 2020) as well as two baseline models, one from RobustBench (Standard RB) and one trained by us (Baseline). All blue dots represent adversarially trained networks for the purpose of clarity we marked three popular models from Carmon et al. (2019), Wang et al. (2020) and Hendrycks, Lee, and Mazeika (2019) by name.

Bench (Croce et al. 2020) exhibits less aliasing than the one trained by us. Unfortunately, there is no further information about the exact training schedule from RobustBench, such that we can not make any assumptions on the interplay between model hyperparameters and aliasing.

**Preact ResNet-18**  We conducted the same measurements for the Preact ResNet-18 as we did for the WRN-28-10 and used the same training procedure described in the appendix. Additionally, we needed to account for one more layer with two additional down-sampling operations.

The overall results, presented in Figure 5, are similar to the ones for the WRN-28-10 networks, most adversarially trained networks exhibit much less aliasing and higher robustness than conventionally trained ones. Yet, the additional down-sampling layer allows one further observation. While the absolute aliasing metric is overall lower, the robust networks reduce the aliasing predominantly in the earlier layers, the second and third layers. The aliasing in the fourth layer of adversarially robust models is not significantly different from the aliasing in conventionally trained models in the same layer.

## Discussion

Our experiments reveal that common CNNs fail to subsample their feature maps in a Nyquist-Shannon conform way and consequently introduce aliasing artifacts. Further, we can give strong evidence that aliasing and adversarial robustness are highly related. All evaluated robust models exhibit significantly less aliasing than standard trained models.

After the application of down-sampling operations in standard CNNs all feature maps suffer from aliasing artifacts occurring due to insufficient sub-sampling.

Adversarially trained networks exhibit significantly less

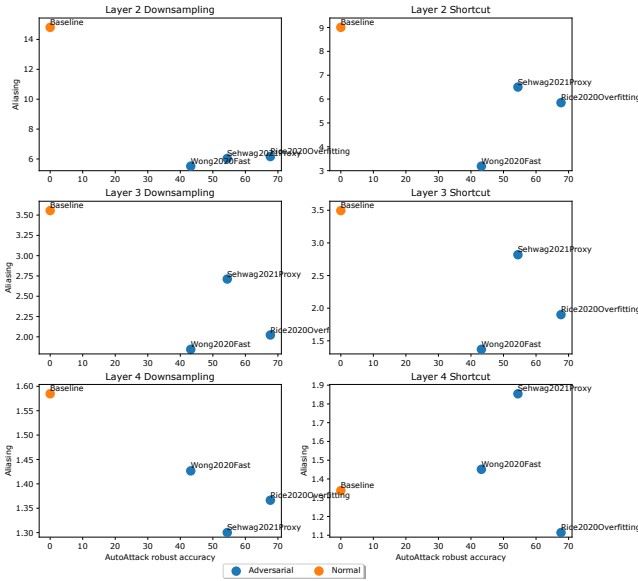

Figure 5: Adversarial robustness versus aliasing exemplary evaluated on different pre-trained Preact ResNet-18 models. The blue dots represent adversarial trained networks, trained with the training schemes of Wong, Rice, and Kolter (2020), Rice, Wong, and Kolter (2020) and Sehwag et al. (2021) provided by RobustBench (Croce et al. 2020). The orange dot is the baseline, trained by us without adversarial training.

aliasing in their feature maps than standard trained networks with the same architecture. As shown previously this is valid for different model architectures and training schemes, especially in the early layers, closer to the input layer. It raises the question whether models with a low amount of aliasing are necessarily more robust. It further entails the question whether there are additional factors that are relevant in this context such as padding techniques, for example. These aspects will be subject to future research.

## Conclusion

Concluding, we were able to show strong evidence that aliasing and adversarial robustness of CNNs are highly correlated. We hypothesize that aliasing is one of the main underlying factors that lead to the vulnerability of CNNs. Recent methods to increase model robustness rather heal the symptoms of the underlying problem than investigate its origins. To overcome this challenge we might need to start thinking about CNNs in a more signal processing manner and account for basic principles from this field, like the Nyquist-Shannon theorem, which gives us clear instructions on how to prevent aliasing. Still, it is not straightforward to incorporate this knowledge into the architecture and structure of common CNN designs. We aim to give a new and more traditional perspective on CNNs to help improve their performance and reliability to enable their application in real-world use cases.

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

## A1: Downsampling Block Preact ResNet

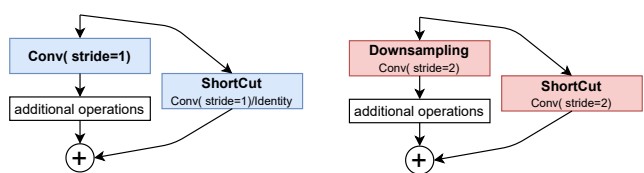

Figure 6: Abstract Illustration of a building block in Preact ResNet-18 and WRN-28-10. The first operation in a block is a convolution. This convolution is executed with a stride of either one or two. For a stride of one (left) the shortcut simply passes the identity of the feature maps forward. If the first convolution is done with a stride of two, the shortcut needs to have a stride of two (right) too, to guarantee that both representations can be added at the end of the building block.

## A2: Training Procedure

The baseline models for the Preact ResNet-18 and the WRN-28-10 are both the same trained with the same schedule. Each model is trained with 200 epochs, a batch size of 512, cross entropy loss and stochastic gradient descent (SGD) with an adaptive learning rate starting by 0.1 and reducing it at 100 and 150 epochs by a factor of 10, a momentum of 0.9 and a weight-decay of 5e-4.