# OpenReview forum: "Aliasing coincides with CNNs vulnerability towards adversarial attacks"
_AAAI.org/2022/Workshop/AdvML — AAAI-22 AdvML Workshop ShortPaper_

### Official Review · Reviewer_HPbf · 2021-12-01
**Review of Paper 6**

**Rating:** 7
**Confidence:** 4

**Review:**

This paper highlights an interesting observation that robust models have learned to downsample more accurately and suffer significantly less from aliasing than baseline models. SOTA models are considered using RobustBench, while visualization results seem promising.

Although showing a robust model suffers less from aliasing is cool, it would be much more intriguing to show vice versa, i.e., a model that suffers less from aliasing is more robust. This cause-effect relationship is not always reversible, for example, a robust model has semantic input gradients, but a model with semantic input gradients (e.g., a generative one learned by score matching) may not be robust.

---

### Decision · Program_Chairs · 2021-12-01

**Decision:**

Accept (Short Paper)

**Comment:**

The reviewer agrees to accept this paper.